# Impact of Physical Activity on Autonomy and Quality of Life in Individuals with Down Syndrome: A Systematic Review

**DOI:** 10.3390/healthcare12020181

**Published:** 2024-01-11

**Authors:** Antonio Muñoz-Llerena, Laura Ladrón-de-Guevara, Daniel Medina-Rebollo, Virginia Alcaraz-Rodríguez

**Affiliations:** 1Department of Physical Education and Sports, University of Seville, 41013 Seville, Spain; amllerena@us.es; 2Research Group “Social Inclusion, Physical Education and Sport, and European Policies in Research - INEFYD” (HUM-1061), University of Seville, 41013 Seville, Spain; lguevara@ceu.es (L.L.-d.-G.); dmedina@ceu.es (D.M.-R.); 3Department of Physical Education and Sport, Centro de Estudios Universitarios Cardenal Spínola CEU, 41930 Seville, Spain; 4Department of Sports and Computer Sciences, Universidad Pablo de Olavide, 41013 Seville, Spain

**Keywords:** Down syndrome, well-being, adapted physical activity, adapted sports, health, intellectual disability

## Abstract

Down syndrome (DS) is the most common genetic alteration in humans, resulting from the trisomy of chromosome 21. Individuals with DS are characterized by physical traits and limitations related to intellectual functioning and the development of motor skills. People with DS tend to have lower levels of physical activity (PA) than the general population, despite its benefits for health and quality of life, which could be caused by barriers such as the lack of adapted programs or knowledge on how to adapt them. Therefore, this systematic review aimed to examine the impact of physical activity or sports programs on autonomy and quality of life in individuals with DS. Preferred Reporting Items for Systematic Reviews and Meta-Analyses guidelines were followed to search four databases (Scopus, Web of Science, PubMed, and SportDiscus), adhering to the population, intervention, comparison, and outcome strategy. A total of 13 studies were selected that followed different training programs (combined training, adapted football, technology-based, pulmonary training, gymnastics and dance, water-based, and whole body vibration). Most of the interventions provided benefits for autonomy or quality of life (physical, psychological, cognitive, emotional, and social) in people with DS. In conclusion, the benefits of physical activity and sports programs adapted to people with DS are positive.

## 1. Introduction

### 1.1. Epidemiology and General Characteristics of Down Syndrome

Down syndrome (DS) is a common genetic alteration resulting from the trisomy of chromosome 21, causing intellectual disability (ID) [1]. Its incidence is approximately 1 in every 700–1200 births [2], with prenatal diagnosis reducing the number of DS births [3,4,5]. Today, people with DS live longer due to advances in medicine and multidisciplinary care, including adapted physical activity programs [6]. Individuals with DS exhibit physical traits and limitations related to intellectual functioning, facial features, hyperlaxity, hypotonia, and other alterations [7,8]. They may have deficiencies in the development of gross motor skills linked to low muscle tone and ligamentous hyperlaxity [9]. DS contributes to delays in the acquisition of motor patterns and the development of atypical patterns [2,10].

### 1.2. Physical Activity and People with Down Syndrome

Physical activity, understood as an activity which involves people moving, acting, and performing within culturally specific spaces, and contexts, and influenced by a unique array of interests, emotions, ideas, instructions, and relationships [11], is essential for overall well-being and quality of life. However, people with DS tend to be less active due to physical limitations, lack of opportunities, and unawareness of the benefits of physical activity [12]. Physical activity programs can enhance cardiovascular health, muscle strength, motor coordination, balance, and cognitive, emotional, and social skills [13,14]. Despite these benefits, barriers such as a lack of specific programs, lack of knowledge on adapting physical activities, and negative attitudes hinder the participation of DS individuals [15]. Therefore, it is necessary to have physical activity programs specifically designed for people with DS [13,15], promoting inclusion and active participation [16].

### 1.3. Autonomy and Quality of Life

Autonomy and quality of life are complex and multidimensional concepts, whose importance for the general population (and, therefore, for people with DS) must be highlighted. On one hand, autonomy is understood as the ability to make decisions about one’s own life and live based on those decisions [17,18], including the perception that one is the source of their own actions [18]. Autonomy also includes aspects such as self-care and mobility [19].

On the other hand, following the model developed by Schalock and Verdugo [20], which is the most used in the field of ID, quality of life refers to the degree to which people value their own life experiences, presenting a series of dimensions that contribute to a full and interconnected life within physical, social, and cultural contexts. In short, it is a desired state of personal well-being that: (1) is multidimensional; (2) has universal and culture-bound properties; (3) has objective and subjective components; (4) is influenced by personal characteristics and environmental factors, and (5) is dynamic and changes over time [21,22,23,24,25]. There are eight dimensions of quality of life established in the model of Schalock and Verdugo [20]: emotional well-being, interpersonal relationships, material well-being, personal development, physical well-being, self-determination, social inclusion, and rights.

Both concepts (autonomy and quality of life) are interrelated. Following Schalock and Verdugo [20], autonomy is an intrinsic aspect of the dimension of self-determination within quality of life. However, in this work, it has been highlighted due to its special relevance for the autonomous and daily performance of people with DS.

### 1.4. Impact of Physical Activity on Quality of Life in People with Down Syndrome

Adapted physical activity programs for people with DS are beneficial in multiple ways. Regular physical activity enhances muscle tone, strength, balance, and motor skills, promoting autonomy and independence in daily activities [26]. It also improves cardiovascular health, body composition, and psychological well-being [27]. Cognitive benefits include enhanced academic performance, cognitive function, attention, concentration, and memory [28,29]. Furthermore, physical activity fosters social integration, quality of life, development of social skills, peer interaction, and friendship bonds [30]. It also boosts self-esteem, self-confidence, and the enjoyment of leisure time [31].

In summary, adapted physical activity programs can have a positive impact on the autonomy and physical, cognitive, social, and emotional health and quality of life of people with DS. These programs should be designed considering the individual characteristics of each person, adapting exercises and activities to their needs and capabilities. In addition, it is necessary to promote education and awareness about the importance of physical activity in this population, both in the family and in academic and professional fields. By providing opportunities and support for participation in physical activities, we can contribute to improving the quality of life and well-being of individuals with DS. However, there has been no specific systematic literature review on the impact of different physical activity and sports programs on autonomy and quality of life (in all its dimensions: physical, cognitive, social, and emotional) in people with DS in recent years. Therefore, the objective of this systematic review was to answer the following research questions:Is physical activity or sports an effective strategy to increase autonomy and to improve physical, cognitive, emotional, and social quality of life in people with DS?What is the methodological quality of the investigations that have studied this topic?Is there any type of physical activity or sports intervention that should be encouraged and prescribed to this population to enhance autonomy and quality of life?

## 2. Materials and Methods

### 2.1. Data Source and Search Protocol

This study followed the reference guidelines set out in the PRISMA statement for systematic reviews and meta-analyses [32]. The review protocol was registered in the PROSPERO database (registration number CRD42023486634).

The search was conducted using four different databases: SCOPUS, Web of Science, MEDLINE (via PubMed), and SportDiscus. The final search was conducted on 23 October 2023. The search was designed with reference to the PICO acronym [33], in which the population included any person with DS, who had an intervention with physical activity and/or sport, and that the variables included in the studies were related to quality of life, autonomy, or the well-being of the participants. The term used for the population was DS, for the intervention sport and/or physical activity, and for the results, terms such as autonomy, quality of life, or well-being, both in English and Spanish. Both controlled and uncontrolled descriptors were applied and combined using Boolean operators as detailed in the following search protocol (Table 1).

### 2.2. Study Selection Criteria

Once the search of the four databases was completed, duplicate articles were removed. Second, an initial screening was carried out with reference to the title and abstract. After obtaining the relevant articles for review, the full texts were read to identify those that met the inclusion criteria and were included in the systematic review. In addition, two researchers independently and in duplicate carried out the first screening, with differences resolved through discussion or inclusion by a third author.

The inclusion criteria were as follows: (1) studies that included subjects with DS; (2) empirical studies that used physical activity or sport as a form of intervention; (3) studies that addressed the variables of quality of life, well-being, or autonomy (at least one of them); (4) articles published in the last 15 years (2008–2023); and (5) in English or Spanish. On the other hand, studies were not included if: (1) there was no planned intervention; (2) the studies were reviews, communications, and/or press articles; and (3) the full text was not available for access.

### 2.3. Methodological Quality Assessment

To evaluate the methodological level of the studies, the Evidence Project tool was applied [34], which allows evaluation of the risk of bias in both randomized and non-randomized studies. This tool is composed of eight items, which were answered yes or no depending on whether they were met in the study: (1) cohort, (2) control or comparison group, (3) pre-post intervention data, (4) random assignment of participants to the intervention, (5) random selection of participants for assessment, (6) follow-up rate of 80% or more, (7) comparison groups equivalent on sociodemographics, and (8) comparison groups equivalent at baseline on outcome measures. For some of them, there were additional options such as “not applicable,” if the criterion does not apply given the study design, or “not reported,” if the fulfillment of the criterion could not be determined by the information presented in the study.

Following the criteria of the Evidence Project, for a study to be selected for this review, it had to comply with at least one of the first three items, which were related to the study design.

### 2.4. Use of Artificial Intelligence

Artificial intelligence (AI) has been utilized in this manuscript to translate and proofread the final manuscript before sending to a native English translator, who checked and proofread the final text. Specifically, Bing Chat was used to translate from Spanish to American English and proofread the whole document, using the following prompt: “Topic: Impact of physical activity programs in people with Down Syndrome. Context: Systematic Review for scientific publication in the journal Healthcare from MDPI. Requirement: Translate to American English, proofread and make the following text cohesive. Language: Academic. Tone: Formal”. Additionally, a second proofread was carried out with Paperpal Prime (version 2.73.0) through its Microsoft Word (Office 365) add-in.

No data or information presented in this work have been created or generated by AI.

## 3. Results

Figure 1 shows the PRISMA flow diagram. A total of 137 documents were obtained from the database search. After eliminating 28 duplicates, 109 articles were screened, of which 30 were evaluated. Finally, 13 studies were selected; the remaining 17 were excluded for various reasons: (1) articles that did not address empirical research that used physical activity or sport as a form of intervention (n = 5); (2) articles that did not address research with the DS population (n = 10); (3) articles that did not address quality of life, well-being, and autonomy (n = 1); and (4) articles where the full text was not available (n = 1).

### 3.1. Methodological Quality Analysis

The analysis based on the Evidence Project is presented in Table 2. The 13 studies complied with at least one of the three criteria related to the study design; therefore, none were excluded from the review. Items 1 and 3 were met by 92.3% of the studies; Items 6, 2, and 8 were met by 84.6%, 61.5%, and 53.8%, respectively. The remaining items were met in less than half of the selected investigations.

### 3.2. Characteristics of the Intervention

Table 3 describes the characteristics of the interventions conducted in each study. Different training methods or physical activity practices were used in the selected studies. Five of them used combined training programs in which different exercises focused on the development of different basic physical qualities were performed (i.e., strength exercises, cardiorespiratory endurance, flexibility, etc.). Two of the interventions used adapted physical activity programs [44,47], while the remaining three used three different types of combined training: Kashi practices, a combination of cardiovascular training, strength, balance, and flexibility [38]; muscle resistance training and assisted cycling [42]; and fitness training through the DSFit group exercise program [45]. On the other hand, two of the studies carried out interventions through adapted soccer training [43,46]. The rest of the articles included in the review used different types of physical interventions: exergaming, through an exercise program based on the Wii device [39]; video-modelling, prompting, and behavior-specific praise [37]; pulmonary training [41]; gymnastics and dance [36]; water-based exercise [40]; and whole body vibration training [35].

### 3.3. Characteristics of the Sample

A description of the sample is provided in Table 3. Of the thirteen selected studies, three included children aged between 3 and 11 years [36,37,41], six included teenagers aged 12 to 17 years [35,36,41,45,46,47], and seven included adults over 18 years of age [38,39,40,42,43,44,47]. Several of them included two different age ranges: children and teenagers [36,41] or teenagers and adults [47].

### 3.4. Impact of the Intervention

The results obtained in the different studies, as well as the variables analyzed, are included in Table 3. Below, they are broken down into results related to the autonomy, quality of life, and well-being of the participants.

#### 3.4.1. Autonomy

Six articles analyzed the impact of physical activity on the autonomy of the participants: two in children and adolescents [45,46] and three in adults [38,39,42]. Specifically, five aspects related to autonomy were addressed: cognitive function, functional mobility, motor proficiency, adherence, and independence.


**Cognitive Function**


Three studies [42,45,46] analyzed the effects of interventions on the cognitive function of a total of 51 participants (37 children and adolescents, 14 adults). More specifically, these studies addressed two specific aspects of cognitive function: the behavior of the participants and their cognitive planning and decision making.


**Functional Mobility and Motor Proficiency**


A total of 53 adults were included in two studies [38,39] that analyzed the impact of two physical activity programs on functional mobility and motor proficiency. 


**Adherence and Independence**


The impact of the intervention on adherence to physical activity practice in two participants who repeated the intervention and the parents’ perception of the changes in the 12 participants’ independence was analyzed by Hojlo et al. [45].

#### 3.4.2. Quality of Life

The thirteen selected studies analyzed the impact of the interventions on the quality of life of the participants. Six studies were conducted in a young population [35,36,37,41,45,46], six in an adult population [38,39,40,42,43,44], and one in both types of populations [47]. These thirteen studies addressed quality of life on its different dimensions: physical, psychological, cognitive, emotional, and social well-being.


**Physical Wellbeing**


All of the studies analyzed some of the components of physical well-being. More specifically, the effects of the interventions on physical fitness and psychomotor skills, anthropometric measures, healthy habits and the use of free time, posture and baropodometry, amount of moderate-to-vigorous physical activity (MVPA), and, finally, the health, quality of life, and general well-being of the participants were analyzed in these studies.


*Physical Fitness and Psychomotor Skills*


Regarding physical fitness and psychomotor skills, nine investigations analyzed a total of 85 children and adolescents [36,41,45,46] and 76 adults [38,39,40,42]. Benavides Pando et al. [47] included a final sample of 18 adolescent and adult participants (12–27 years), but they do not specify in their study how many correspond to each age group. Within this dimension of physical well-being, the variables analyzed in the nine studies were strength, balance, aerobic endurance, cardiorespiratory fitness, speed, agility, coordination, flexibility and mobility, and pulmonary function.


*Anthropometric Measures*


Regarding anthropometric measures, five studies analyzed 81 children and adolescents [35,41,45] and 39 adults [39,40]. This dimension includes the variables of height, weight, BMI, waist circumference, body composition, and the ratio of the upper to lower chest wall.


*Healthy Habits and Leisure Time Use*


Two studies evaluated the influence of interventions on healthy habits and leisure time use in 12 children and adolescents [45] and 14 adults [40].


*Posture and Baropodometry*


Di Fabrizio et al. [44] were the only ones who addressed body posture and baropodometry of 10 adults.


*Amount of Moderate to Vigorous Physical Activity*


Only Adamo et al. [37] studied the impact of the intervention program on the amount of MVPA in three preschool children.


*General Health, Quality of Life and Wellbeing*


Three studies analyzed the general health, quality of life, and well-being of 12 children and adolescents [45] and 53 adults [40,43].


**Psychological, Cognitive, and Emotional Wellbeing**


Four of the 13 studies addressed the psychological and cognitive well-being of participants [40,43,45,46]. The specific variables analyzed within this dimension of quality of life include self-esteem, self-determination, emotional state and mood, satisfaction, and personal development.


*Self-Esteem and Self-Determination*


Three studies analyzed the impact of the interventions on self-esteem and self-determination in 12 children and adolescents [45] and 53 adults [40,43].


*Emotional State and Mood*


Emotional state and mood were analyzed in 37 young people [45,46] and 39 adults [43] in three different studies.


*Personal Satisfaction and Development*


Similar to the variables of self-esteem and self-determination, three studies analyzed the impact of the interventions on satisfaction and personal development in 12 children and adolescents [45] and 53 adults [40,43].


**Social Wellbeing**


Only 3 of the 13 articles studied the social well-being of the participants [43,45,46]. More specifically, these investigations analyzed the variables of interpersonal relationships, social inclusion, and social rights.


*Interpersonal Relationships*


The three previously mentioned studies evaluated aspects related to interpersonal relationships in 37 young people [45,46] and 39 adults [43].


*Social Inclusion and Rights*


Camacho et al. [43] were the only ones who addressed the variables of social inclusion and social rights.

## 4. Discussion

The objective of this review was to summarize, critically evaluate, and integrate existing scientific knowledge regarding the impact of physical activity and sports on the autonomy and physical, cognitive, social, and emotional quality of life of people with DS. A total of 13 studies of adequate methodological quality were found, allowing the creation of a useful knowledge base for future researchers and professionals in this field. These studies have used various types of interventions based on physical activity and sports, with combined training (i.e., training focused on improving various basic physical qualities) predominating among them.

While some types of interventions, such as whole body vibration or aquatic activities, did not report any type of improvement or positive change in the participants, the majority of them did produce an evolution in at least one component of autonomy or physical, cognitive, social, or emotional quality of life, both in adults and young people. Special mention should be made of interventions focused on technological elements (e.g., exergames, video simulation), which seem to be valuable tools for performing physical activity and improving motor skills with greater motivation, coinciding with what other authors have previously stated [48,49].

The results of this work align perfectly with the quality of life model established by Schalock and Verdugo [20]. Thus, the conclusions reached by the different studies align with the dimensions established in this model (i.e., emotional well-being, interpersonal relationships, material well-being, personal development, physical well-being, self-determination, social inclusion, and rights). The interpretations of the results on the effects of interventions on autonomy and quality of life, including the latter’s physical, psychological, cognitive, emotional, and social well-being, are shown below.

### 4.1. Effects of Interventions on Autonomy

The results obtained showed that the analyzed research addressed five aspects of autonomy: cognitive function (i.e., the behavior of the participants and their cognitive planning and decision making), functional mobility, motor proficiency, adherence to physical activity practice, and independence.


**Cognitive Function**


In relation to behavior, Hojlo et al. [45] showed a decrease in specific symptoms of hyperactivity and impulsivity in almost all participants in the two cohorts of their study, although there were no positive changes in more general behavioral problems (e.g., irritability, agitation, stereotypic behavior, lethargy, and social withdrawal). In contrast, Perić et al. [46] achieved with their intervention a significant improvement in four psychosocial variables in the experimental group: aggression, attention disorders, anxiety and depression, and social problems, which was not reflected in the control group, where there were no significant changes in any of the four variables. Finally, Ringenbach et al. [42] showed an improvement close to significance for the two experimental groups (resistance training and assisted cycling training) in terms of inhibition control, although the statistical results were not significant.

Regarding cognitive planning and decision making, the qualitative results obtained by Hojlo et al. [45] on goal setting showed that, thanks to the intervention carried out, both participants and parents or tutors began to set goals related to a greater amount of physical activity practice and being healthier. On the other hand, Ringenbach et al. [42] obtained significant improvements in the cognitive planning of participants who participated in the experimental group of assisted cycling training and in the control group (who played board games), as well as a decrease in this variable in the experimental group of resistance training.


**Functional Mobility and Motor Proficiency**


On the one hand, Kashi et al. [38] obtained significant improvements in the reaction time of participants in their intervention through Kashi practices. On the other hand, the participants in the experimental group in the study by Silva et al. [39] showed significantly improved functional mobility and response speed, variables that did not show significant changes in the control group.


**Adherence**


Regarding the impact of the intervention on adherence to physical activity practice, Hojlo et al. [45] showed that both participants who repeated the intervention in the second iteration increased their practice of physical activity and sports in their daily life: the first was through daily practice of the DSFit program exercises; the second was by going from not wanting to practice physical activity to signing up for the local Special Olympics basketball team and starting to regularly attend her school gym.


**Independence**


Within the qualitative results of the study by Hojlo et al. [45], the parents of the 12 adolescent participants considered that the DSFit program favored the latter’s independence and facilitated them to carry out new routines on their own.

Maintaining appropriate levels of muscle strength makes people with DS healthier, helping them live independently and autonomously [38]. However, this population is less active than the general population [10], and, like any person without physical activity habits, they see their levels of sedentary behavior tremendously increased when they reach adolescence, which causes them to stop being able to perform the activities that are carried out in physical education sessions, training, or in tasks of their daily life. This sedentary life exacerbates their health problems, leading to a loss of autonomy when reaching adulthood [15,47]. Therefore, it is essential to know how to design and adapt physical activities for these people, which must be motivating so that they perceive them as leisure and playful activities and must be adapted to the capabilities. If not, they will cause rejection and, therefore, lack of adherence to the activities. If the activity is not an obligation, it can be maintained over time and produce positive effects on the participants [50], influencing their daily personal performance and quality of life, producing an increase in adherence to the practice of daily physical activity, especially in activities with family members or close people. Therefore, it is recommended that any type of adapted physical activity be carried out, which in the company of family members would be even more beneficial. These results can be attributed to the effects of physical activity on the self-realization and independence of people with DS and those in their close environment [51].

People with DS may need adaptations in activities to promote their involvement. Therefore, organizing adapted activities and providing them with clear and direct instructions can contribute to this type of population being able to participate successfully [37]. In addition to the above, improving inhibition and cognitive planning is essential for carrying out daily life activities, independence, and employment, and one way to favor this aspect is the use of moderate-intensity physical activity [42].

The results of this review support the positive impact of exercise interventions on individuals in relation to the daily life activities of people with DS, developing greater autonomy and independence. This development was due to improved behavior, derived from the reduction in hyperactivity, impulsivity, and various psychosocial variables such as aggression, attention disorders, and social problems; the development of the ability to plan and make decisions; and an improvement in reaction speed and functional mobility. These results have also been reported in other studies [51,52,53,54,55].

### 4.2. Effects of Interventions on Physical Wellbeing

The included studies addressed the effects of interventions on physical fitness and psychomotor skills (i.e., strength, balance, aerobic endurance, cardiorespiratory fitness, speed, agility, coordination, flexibility and mobility, and pulmonary function), anthropometric measures (i.e., height, weight, BMI, waist circumference, body composition, and the ratio of the upper to lower chest wall), healthy habits and the use of free time, posture and baropodometry, amount of MVPA, and, finally, the health, quality of life, and general well-being.


**Physical Fitness and Psychomotor Skills**


Regarding strength, seven previous studies analyzed the impact of the interventions on the development of strength and power in the lower limbs, upper limbs, core, and respiratory muscles. For the lower limbs, three studies obtained statistically significant improvements in jump length in the experimental group [38,39,47], an improvement that was not observed in the control groups in the studies by Kashi et al. [38] and Silva et al. [39]. In contrast, Ayán Pérez et al. [40] did not observe statistically significant changes in jump length during their intervention. In the case of Hojlo et al. [45] and Moraru et al. [36], improvements were observed in the strength of the lower body in all participants, but they were not statistically significant.

For the upper limbs, Kashi et al. [38] obtained significant improvements in handgrip strength and upper-body muscle endurance in the experimental group, whereas Silva et al. [39] also obtained significant improvements in handgrip strength in the experimental and control groups. On the other hand, the participants of the studies by Ayán Pérez et al. [40] and Benavides Pando et al. [47] did not have significant changes in handgrip strength; the same happened with muscle endurance in the research by Silva et al. [39]. Finally, Hojlo et al. [45] reported improvements in muscle endurance in participants that were not significant.

For core strength, Kashi et al. [38] achieved significant improvements in both trunk strength and core muscle endurance; the same occurred with respect to muscle endurance in two other studies [39,47]. Hojlo et al. [45] and Moraru et al. [36] also obtained improvements in muscle endurance, although these were not statistically significant.

Finally, only one study analyzed the strength of the respiratory muscles [41] by measuring the maximum inspiratory and expiratory pressures, obtaining significant improvements in both experimental groups and significant differences between the experimental groups and the control group and between the two experimental groups (in favor of the group that performed proprioceptive neuromuscular facilitation).

Regarding balance, five studies addressed this variable using different tests. On the one hand, three of them showed positive results: Kashi et al. [38], who obtained significant improvements in the experimental group that were not later reflected in the control group, and Hojlo et al. [45] and Moraru et al. [36], who observed non-significant improvements. In contrast, two other studies [39,47] did not observe significant changes in the balance of the experimental group.

Five other articles assessed the effects of interventions on the aerobic endurance and cardiorespiratory fitness of the participants. Three studies obtained significant improvements in the experimental groups and significant differences between groups in aerobic capacity [39,41] and heart rate [42]. Two other studies showed positive changes in aerobic capacity [45] and cardiorespiratory fitness [40], although they were not significant.

Five investigations analyzed the effects on the speed, agility, and coordination of the participants. Three of these [38,39,47] obtained significant improvements in speed, with differences between the experimental and control groups [38,39]. Furthermore, three other studies showed improvements in the agility of the subjects in the experimental groups, two of which were statistically significant [38,39], and the other did not [40]. In contrast, Silva et al. [39] observed significant differences between the experimental and control groups in upper-limb coordination. These results were similar to those of Benavides Pando et al. [47], although the latter only observed improvements in the right upper limb and not in the left limb. Finally, Perić et al. [46] achieved significant improvements in specific soccer coordination in the experimental group (i.e., ball control with dribbling).

Regarding flexibility and mobility, three studies obtained significant improvements in the experimental group [39,47] and non-significant improvements [45] in lower body flexibility. Hojlo et al. [45] also observed non-significant improvements in the flexibility of the upper body. In addition, a fourth study [36] showed improvements in spinal mobility in all three participants.

Finally, pulmonary function (vital capacity, forced expiratory volume in 1 s, peak expiratory flow rate, and maximum voluntary ventilation) was analyzed by Mohamed et al. [41]. The results showed significant improvements in all variables of pulmonary function in both experimental groups, as well as in vital capacity and peak expiratory flow rate in the control group. In addition, there were significant differences between the experimental and control groups, and between the two experimental groups (in favor of the group that performed proprioceptive neuromuscular facilitation).


**Anthropometric Measures**


Regarding height, weight, and BMI, none of the five studies reported significant improvements after the intervention [35,39,40,41,45]. The same was true for waist circumference in the study by Ayán Pérez et al. [40], although Silva et al. [39] showed significant improvements in this variable. Of the three studies that analyzed body composition, only two achieved positive changes in the experimental group: one in the percentage of fat mass in the upper limbs [35] and the other in visceral fat [39]. However, none of these changes has been observed with respect to muscle mass [35,39,40]. Finally, only Mohamed et al. [41] evaluated the ratio of the upper to lower chest wall and obtained results similar to the rest of the variables in this study: significant improvements in both experimental groups and in the control group, as well as significant differences between both experimental groups and the control and between the two experimental groups (in favor of the group that performed proprioceptive neuromuscular facilitation).


**Healthy Habits and Leisure Time Use**


In adults, no changes were observed in daily habits or behaviors during leisure time, and the qualitative results of Hojlo et al. [45] showed that, in young people, exercise and physical activity became an important part of daily and family life. In fact, for one of the girls who repeated the two interventions, having participated in the study allowed her to gain sufficient confidence to sign up for a local Special Olympics basketball team and to regularly attend the school gym.


**Posture and Baropodometry**


Di Fabrizio et al. [44] reported results which indicated a better distribution of load on both feet and a reduction in podalic overload points, aspects that were related in seven of the participants with an improvement in the angular position of the hip, knee, ankle, and foot joints. These postural improvements were due to lesser anteversion of the pelvis, reduction in knee valgus and hindfoot, and elevation of the plantar arch.


**Amount of Moderate to Vigorous Physical Activity**


Adamo et al. [37] showed that all three participants increased their MVPA during the intervention and that, when it ended, the MVPA decreased. However, in the second baseline condition of their study, the MVPA did not decrease to the level of the first one. In fact, during the period between the first and second rounds of intervention, the participants carried out some of the activities shown in the videos during the intervention without watching them.


**General Health, Quality of Life and Wellbeing**


Camacho et al. [43] achieved results that showed significant differences between male and female participants in relation to the quality of life index (in favor of men) in the questionnaires carried out by the informants (e.g., parents, tutors, teachers), as well as a significantly higher quality of life index in the group of athletes compared to the non-athletes (perceived by the participants with DS) and a higher perception of quality of life in the participants than in the informants. However, they also found that the participants did not perceive these significant differences in the quality of life index between men and women, and that the informants did not consider that there were significant differences between athletes and non-athletes.

On the other hand, Ayán Pérez et al. [40] did not find significant changes in the general health of the participants. However, Hojlo et al. [45], in their qualitative results, observed that there was a positive change in the perception of participants and parents related to the importance of maintaining a healthy lifestyle.

The main results on physical well-being showed an increase in physical fitness and psychomotor skills, and specifically in various physical variables (e.g., strength, balance, aerobic endurance, cardiorespiratory fitness, speed, agility, coordination, flexibility, mobility, and pulmonary function) in most of the interventions analyzed. Similar results have previously been reported in the scientific literature on physical activity in DS [31,52,53,54]. However, it is necessary to consider some aspects when implementing physical activity programs in this population. People with DS present many alterations associated with their syndrome, such as muscle weakness, hypotonia, ID, growth retardation in motor development, and a low aerobic capacity [53,56]. In this way, the growth retardation has a direct impact on their learning and control of motor skills [38]. Another important aspect to consider is muscle weakness and hypotonia, very disabling characteristics for people with DS. Therefore, designing adapted physical activity programs facilitates their physical performance and the performance of daily life activities, also improving their health and quality of life [9,16,38,40,57].

Similarly, a greater amount of MVPA can contribute to maintaining cognitive function (e.g., attention, memory) and preventing or delaying the onset of diseases such as Alzheimer’s in people with DS [58]. In fact, concern for one’s own physical well-being and health is a determining factor in the quality of life in older people who have ID [43].

At the anthropometric level, only some of the analyzed studies showed improvements in waist circumference, fat mass, and the upper-to-lower chest wall ratio. These improvements (and their absence in other variables, such as height, weight, or BMI) are similar to those presented in previous studies [56]. This is consistent with the fact that exercise itself does not lead to significant changes in anthropometric parameters or body weight, although aerobic exercise could produce positive changes in factors related to obesity, and especially aerobic interval training [59].

The physical well-being of people with DS decreases with age, given the accelerated aging that characterizes this population, and that implies experiencing a physical state typical of older people [43]. Therefore, it is essential to design specifically adapted physical activity programs to improve physical well-being and prevent the possible consequences of premature aging and a sedentary lifestyle in this population.

### 4.3. Effects of Interventions on Psychological, Cognitive, and Emotional Wellbeing

The literature review showed that analyzed research evaluated the effects of physical activity on self-esteem, self-determination, emotional state and mood, satisfaction, and personal development.


**Self-Esteem and Self-Determination**


Hojlo et al. obtained promising qualitative results regarding self-esteem [45]. In the interviews, it was mentioned that one of the participants did not practice sports before starting the intervention due to a lack of confidence, and after finishing she began to trust herself, signing up for a local Special Olympics basketball team and started to regularly attend the school gym. In contrast, Pérez et al. [40] did not find significant changes in the self-esteem of the participants.

Self-determination was analyzed by Camacho et al. [43], who found significant differences between athletes and non-athletes (in favor of the former) according to the perception of the participants, as well as an absence of differences between sexes (according to participants and informants) and between athletes and non-athletes (only informants). There were also significant differences in this variable between participants and informants (in favor of the participants).


**Emotional State and Mood**


In adults, Camacho et al. [43] evaluated the differences in emotional state and found that men had a significantly better emotional state than women (observed by participants and informants). According to what was observed by the participants, the same thing happened in athletes with respect to non-athletes (while, for the informants, there were no significant differences), and there were no significant differences between participants and informants in the perception of this variable. In young people, a reduction in anxiety and depression was observed in the experimental group, which in the case of Perić et al. [46] was significant and did not occur in the control group. Hojlo et al. [45] found that this reduction occurred in the majority of participants from both cohorts but was not significant; they also found that both participants and parents increased their interest in practicing exercise as a mechanism for improving mental health.


**Personal Satisfaction and Development**


In the research by Camacho et al. [43], participants perceived significantly higher personal development values in men than in women and in athletes than in non-athletes, differences that were not significant in the perception of the informants, and there were also statistical differences between participants and informants regarding this variable. However, Pérez et al. [40] did not find significant changes in participant satisfaction after the intervention. Finally, the interviews conducted by Hojlo et al. [45] showed that the intervention served to increase the satisfaction of the participants, and that the key aspects for both participants and parents to feel satisfied were the fun, positive, and social atmosphere of the program; being able to meet other families in the same situation; and practicing walking exercises, with ball throwing, or with music and dance. The only aspect they mentioned that hindered satisfaction was the standing and waiting exercises.

Among the analyzed studies, several stand out for the treatment of psychological, cognitive, and emotional well-being, obtaining results such as increases in self-esteem and self-determination and a better emotional state and mood (especially in men and in relation to anxiety and depression). While there are few contributions from the literature regarding this theme, some studies [51,53] have obtained similar results on the emotional state in this type of population. There were also investigations in this review that revealed that men showed greater personal development than women, and that there are different key aspects that favor the satisfaction of this population in the practice of physical activity (e.g., a positive social climate, being relational with other families and peers, and practicing exercises that include movement and music). Similar conclusions have been reached in other studies in the scientific literature related to this theme [52,53,54]. It is important to highlight that a positive emotional state produces a beneficial effect on the well-being of any person, and especially in people with ID, as it can help them deal with the barriers they may encounter in society [43].

### 4.4. Effects of Interventions on Social Wellbeing

Within social wellbeing, the reviewed research analyzed interpersonal relationships, social inclusion, and social rights.


**Interpersonal Relationships**


In children and adolescents, the interviews conducted by Hojlo et al. [45] showed that what the participants enjoyed the most was learning new exercises, being together with other children, and being able to relate to them in the program. In addition, Perić et al. [46] significantly reduced social problems (e.g., arguing with others, getting angry easily, not being liked by other people, preferring to be alone rather than with others) in the experimental group, a reduction that did not occur in the control group. In adults, the participants in the study by Camacho et al. [43] considered that there were significant differences in interpersonal relationships between athletes and non-athletes (i.e., the group of athletes communicated better with others); these differences were not perceived by the informants. No significant differences were observed in this variable between men and women or between the perceptions of the participants and informants.


**Social Inclusion and Rights**


Participants in the study of Camacho et al. [43] perceived that there were no differences in social inclusion and rights based on sex, although the group of athletes presented significantly higher values than non-athletes. Furthermore, the informants did not perceive differences in social rights according to sex or in any of the variables depending on whether they were athletes; however, they did present significantly higher values in men in social inclusion. Lastly, there were no differences between participants and informants in the values obtained in the variable of social rights, but there were differences in social inclusion (the participants presented significantly higher values).

The results of this review showed that physical activity or sports interventions favor interpersonal relationships and social inclusion, which is similar to previous research [52,53,54]. On a social level, the family is a fundamental agent that must generate favorable contexts for people with (DS) to make future plans [43]. Furthermore, some people with DS participate better in physical activities when family members or other key adults practice with them, suggesting that social interaction is one of the main motivating factors for participation in activities in this population [45]. The sports context is a suitable context for generating these social interactions, as it can promote mutual understanding, cooperation, interpersonal relationships, social inclusion, self-determination, and quality of life, being an ideal vehicle for generating social capital [43,60,61]. In fact, competitive team sports are characterized by intense physical and social contacts, a reflection of current society, and serve to reflect the socialization process of the participants [43]. Therefore, the practice of collective physical and sports activities (and, especially, competitive sports) can be useful tools to improve the social well-being of people with DS.

### 4.5. General Considerations

Well-defined and structured physical activity programs allow for improvements in physical, psychological, emotional, and social well-being, greater motor control, greater autonomy and independence, and performance in functional activities of daily life, avoiding the isolation of people with DS and including them in society [10,13,50], and can be complemented with automated tracking systems of practice or with tools that serve as reminders to maintain adherence to the program [45]. However, although the benefits of practicing physical activity in all dimensions of quality of life are evident, the greatest effects are usually given at the psychosocial level and not at the motor level, since the objective of this type of program is usually to improve general motor coordination to successfully carry out daily life activities, and not specific sports actions [46].

It is also necessary to consider the possible challenges when measuring this type of population, as it is difficult for them to adhere to very strict evaluation parameters, and their lack of motivation or understanding of the instructions during the measurement can affect the measurements made [45].

### 4.6. Limitations

Several limitations should be considered in this review. The first is the non-inclusion of grey literature or institutional reports. The second and main limitation is the limited number of trials and research available on the chosen topic. In addition, the sample size in the different studies was also reduced in most of them, which represents a weakness in the statistical analysis of the results analyzed. Another possible limitation could be the inclusion of studies only in English or Spanish.

At a methodological level, one limitation was the clinical heterogeneity of the selected studies in terms of study designs (i.e., different types of physical activity interventions, inclusion of non-randomized studies), characteristics of the samples (i.e., different sizes, ages, and experiences with physical activity), variables analyzed (i.e., different variables and different measurement instruments in the same variables among studies), and research questions (i.e., different research questions and goals among studies). This clinical heterogeneity meant that the performance of a meta-analysis was not pertinent, following the recommendations of Lensen [62] and Escrig Sos [63].

Despite the above, it is necessary to value the results while interpreting the studies analyzed with caution. This review showed that there is a need to carry out more studies on this topic, with a larger sample size and methodological rigor to support and deepen aspects such as the specificity of the prescription of physical activity in people with DS and the benefits that adapted physical activity can have on the autonomy and quality of life of the participants.

## 5. Conclusions

DS is one of the most common pathologies in our society, which carries different alterations that diminish the quality of life of the people who suffer from it and limit the development of their capacities and abilities. The present study showed that physical activity and physical activity and sport programs are beneficial for improving the parameters that make up the quality of life of a person with DS.

Throughout the developmental process of the population with DS, as well as in their adult life, it is important to avoid a sedentary lifestyle, so the implementation of motivating physical activity programs adapted to their characteristics is important. These programs allow them to develop their capacities to the maximum so that in their future life they have a better quality of life and do not aggravate the health problems associated with this disability. A well-structured and adapted physical activity program can have positive effects on physical, psychological, emotional, and social well-being, as well as promote autonomy and independence and facilitate performance in daily life activities. Thus, this review allows us to discard the idea that this population could not or should not perform physical activity or exercise due to the limitations inherent in DS, such as ID, muscle weakness, hypotonia, congenital heart problems, or low aerobic capacity.

As future lines of research, it is necessary to unify concepts regarding the prescription of physical activity in the population with DS, as well as to carry out more standardized interventions, with greater methodological rigor, less clinical heterogeneity, and a larger sample size to be able to perform deeper statistical analyses. Another relevant aspect that could be addressed in future research is the establishment of guidelines and specific measurement batteries for this population, in order to alleviate the deficiencies that currently exist when measuring people with DS. Finally, the authors of this study encourage other researchers to promote and boost research in the field of physical activity in relation to improving health and quality of life in people with ID and, specifically, DS, considering that they are a vulnerable population group.

In conclusion, the benefits in autonomy and quality of life (at the physical, psychological, cognitive, emotional, and social levels) of physical activity and sports programs adapted to people with DS are positive. The promotion and provision of resources at a personal level and in a close environment (family, educational center, reception center) improves the physical level and generates habits of adherence to activity that can generate a higher quality of life in adulthood. Psychosocial aspects are also essential for the success and better adaptation of people with DS; however, they are less attended to by the investigations collected in the review.

## Figures and Tables

**Figure 1 healthcare-12-00181-f001:**
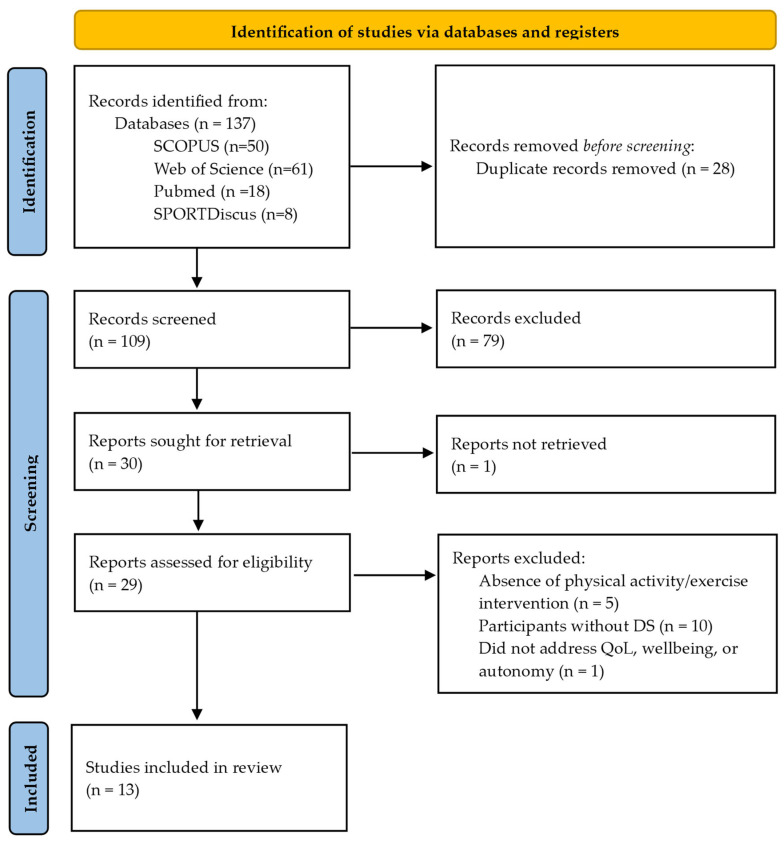
PRISMA flow diagram.

**Table 1 healthcare-12-00181-t001:** Search protocol.

**SCOPUS**		((TITLE-ABS-KEY (“síndrome de down”) OR TITLE-ABS-KEY (“Down Syndrome”))) AND ((TITLE-ABS-KEY (autonomía) OR TITLE-ABS-KEY (autonomy) OR TITLE-ABS-KEY (“calidad de vida”) OR TITLE-ABS-KEY (“quality of life”) OR TITLE-ABS-KEY (wellbeing) OR TITLE-ABS-KEY (bienestar))) AND ((TITLE-ABS-KEY (sport) OR TITLE-ABS-KEY (deporte) OR TITLE-ABS-KEY (“physical activity”) OR TITLE-ABS-KEY (“actividad física”))) AND PUBYEAR > 2007 AND PUBYEAR < 2024 AND (LIMIT-TO (DOCTYPE, “ar”)) AND (LIMIT-TO (LANGUAGE, “English”) OR LIMIT-TO (LANGUAGE, “Spanish”))
**Web of Science**		1. TITLE-ABS-KEY (“syndrome down” OR “sindrome de down”) 2. TITLE-ABS-KEY (autonomía OR autonomy OR “calidad de vida” OR “quality of life” OR bienestar OR wellbeing) 3. TITLE-ABS-KEY (sport OR “physical activity” OR “actividad fisica”) 4. #1 AND #2 AND #3
**MEDLINE (PubMed)**	MeSH Terms and Text Word	((((“Down Syndrome”[Text Word]) OR “Down Syndrome”[MeSH Terms])) AND ((((((autonomy[Text Word]) OR “quality of life”[MeSH Terms]) OR “quality of life”[Text Word]) OR “autonomy, personal”[MeSH Terms]) OR wellbeing[MeSH Terms]) OR wellbeing[Text Word])) AND ((((sport[MeSH Terms]) OR sport[Text Word]) OR “physical activity”[MeSH Terms]) OR “physical activity”[Text Word])
	Title/Abstract	(((((sport[Title/Abstract]) OR (deporte[Title/Abstract])) OR (“actividad física”[Title/Abstract])) OR (“physical activity”[Title/Abstract])) AND ((((((autonomía[Title/Abstract]) OR (autonomy[Title/Abstract])) OR (“calidad de vida”[Title/Abstract])) OR (“quality of life”[Title/Abstract])) OR (wellbeing[Title/Abstract])) OR (bienestar[Title/Abstract]))) AND ((“síndrome de down”[Title/Abstract]) OR (“Down Syndrome”[Title/Abstract]))
**SportDiscus**		S1. SU “Down Syndrome” OR SU “síndrome de down” S2. SU autonomy OR SU autonomía OR SU “calidad de vida” OR SU “quality of life” OR SU wellbeing OR SU bienestar S3. SU deporte OR SU sports OR SU “actividad física” OR SU “physical activity” S4. (SU deporte OR SU sports OR SU “actividad física” OR SU “physical activity”) AND (S1 AND S2 AND S3)

**Table 2 healthcare-12-00181-t002:** Methodological quality analysis.

Study	1. Cohort	2. Control or Comparison Group	3. Pre-post Intervention Data	4. Random Assignment of Participants to the Intervention	5. Random Selection of Participants for Assessment	6. Follow-up Rate of 80% or More	7. Comparison Groups Equivalent on Sociodemographics	8. Comparison Groups Equivalent at Baseline
González-Agüero et al. [35]	Yes	Yes	Yes	Yes	No	Yes	Yes	Yes
Moraru et al. [36]	Yes	No	Yes	NA	No	Yes	NA	NA
Adamo et al. [37]	Yes	Yes	Yes	No	Yes	Yes	NR	Yes
Kashi et al. [38]	Yes	Yes	Yes	Yes	No	Yes	NR	Yes
Silva et al. [39]	Yes	Yes	Yes	Yes	No	Yes	Yes	Yes
Ayán Pérez et al. [40]	Yes	No	Yes	NA	No	Yes	NA	NA
Mohamed et al. [41]	Yes	Yes	Yes	Yes	No	Yes	Yes	Yes
Ringenbach et al. [42]	Yes	Yes	Yes	Yes *	No	Yes	Yes	Yes
Camacho et al. [43]	No	Yes	No	No	No	NA	Yes	NR
Di Fabrizio et al. [44]	Yes	No	Yes	NA	No	Yes	NA	NA
Hojlo et al. [45]	Yes	No	Yes	NA	No	Yes	NA	NA
Perić et al. [46]	Yes	Yes	Yes	Yes	No	Yes	Yes	Yes
Benavides Pando et al. [47]	Yes	No	Yes	NA	No	No	NA	NA

Note. NR, Not reported; NA; Not applicable; *, Randomized order of the activities.

**Table 3 healthcare-12-00181-t003:** Summary of characteristics and results of the included studies.

Study	Sample Characteristics	Intervention	Outcomes (Measurement Tool)	Results
González-Agüero et al. [35]	Sample size (n pre/post; sex) 30/24; 7 female Distribution: group; sex; age (years) IG: n = 11; 3 girls; 15.27 ± 2.57 CG: n = 13; 4 girls; 15.80 ± 3.04	IG: 20 weeks of WBV training 3 training sessions/week in vertical platform Increasing frequency and duration, squat position CG: NR	Height (stadiometer) Weight (stadiometer) BMI (kg/m^2^) Pubertal development (specific observation scale) Fat mass (dual energy X-ray absorptiometry) Lean mass (dual energy X-ray absorptiometry)	IG adherence (%); mean attendance (%): 80%; 80% (final sample) Significant results: IG showed a higher percent declination in fat mass at the upper limbs than CG (*p* < 0.05) Positive (not statistically significant) results: IG showed a tendency toward a higher percent increase in whole body lean body mass (*p* = 0.08) Negative results/no changes: No significant group by time interactions were found for any variable after intervention (all *p* > 0.05)
Moraru et al. [36]	Sample size [n pre/post; sex; age (years)] 3/3; NR; 10–14 years	IG: 8 months of a gymnastics and dance training program Frequency of training NR Warm up: walking, running, specific motor tasks Main part: specific gymnastics and dance exercises (e.g., adapted dance steps, rolls, lateral handsprings, balances, jumps) Cool down: exercises to improve force and resistance, stretching, and breathing	Lower limb strength (timed sit to stand test) Spine mobility (seated forward functional reach test) Core strength (partial sit-up test) Unipodal balance (single leg stance—eyes closed test)	IG adherence (%); mean attendance (%): 100%; NR Positive (not statistically significant) results: Enhancement of the scores of IG in the four outcomes post-intervention
Adamo et al. [37]	Sample size [n pre/post; sex; age (years)] 3/3; 1 female; 4.11 ± 1.03	IG: Packaged intervention including peer video modeling, prompting, and behavior-specific praise from an adult Experimental A-B-A-B withdrawal design: Baseline 1 (7 days), Intervention 1 (9 days), Baseline 2 (4 days), Intervention 2 (7 days) Daily implementation during outdoor period Seven videos (12–24 s in duration) depicting peers doing physical activities and games in the playground An iPad was programed to show participants two videos, they chose whichever they preferred. The video of the activity played, and the participant was instructed to perform it. When finished, a reinforcement video was played and two new options were presented Implementers gave prompts and praises to participants as needed	MVPA (specific observation scale)	IG adherence (%); mean attendance (%): 100%; 98.6% Positive (not statistically significant) results: MVPA increased during the intervention and decreased when the intervention was withdrawn for all participants. MVPA levels in the second baseline did not decrease to the level of the first baseline for any participant
Kashi et al. [38]	Sample size [n pre/post; sex; age (years)] 28/24; no females; 29.19 ± 3.93 Distribution: IG: n = 13 CG: n = 11	IG: 12 weeks of Kashi practices Combination of cardiovascular exercise and strength, balance, and flexibility training Five parts: balance training, strength and power training, muscular endurance and aerobic training, psychomotor skills training, and other exercises (e.g., vibration, dances, and games) 3 training sessions/week, 3 months Incremental length (50–150 min) and intensity (light—difficult) CG: Same conditions (i.e., eating, physical activity, sleeping and participation in the educational program) No exercise or physical activity training	Reaction time (specific subscale of BOTMP) Agility (specific subscale of BOTMP) Balance (specific subscale of BOTMP) Running speed (45 m running test) Power (vertical and long jump) Strength (wrist and trunk dynamometry) Muscular endurance (push-up test, long and sit test)	IG adherence (%); mean attendance (%): 85%; NR Significant results: Pre-test differences between IG and CG were not statistically significant [Wilks’ Lambda = 0.771, f(6,17) = 0.843, *p* = 0.554], but post-test differences were significant [Wilks’ Lambda = 0.428, f(6,17) = 3.8 = 787, *p* = 0.014]. After intervention, there were significant improvements in IG in all the outcomes (*p* < 0.01). CG did not show any significant improvements (*p* > 0.05)
Silva et al. [39]	Sample size [n pre/post; sex; age (years)] 27/25; NR; 18–60 years Distribution: IG: n = 12 CG: n = 13	IG: 2-month Wii-based exercise program included in regular occupational therapy program 22 sessions, 3 sessions/week, 1 h/session 11 individual sessions, 11 paired sessions Individual sessions: balance and isometric strength exercises using Wii Fit Balance Board and different individual games Paired sessions: aerobic endurance exercises using sports-related and dancing games CG: Usual daily activities in the occupational therapy program	Height (stadiometer) Waist circumference (anthropometric tape) Weight (segmental body composition analyzer) BMI (segmental body composition analyzer) Body fat % (segmental body composition analyzer) Visceral fat (segmental body composition analyzer) Muscle mass (segmental body composition analyzer) Coordination (plate tapping test, beanbag overhead throw test) Strength (handgrip test) Running speed and agility (shuttle run test) Balance (flamingo test) Flexibility (sit and reach test) Power (standing broad jump test) Muscular endurance (30 s sit-ups test, bent arm hang test) Aerobic endurance (6-minute walk test) Functional mobility (timed up and go test) Reaction time (specific subscale of BOTMP)	IG adherence (%); mean attendance (%): 93%; NR Significant results: There were significant group by time interactions on flexibility (*p* = 0.027), lower limbs power (*p* = 0.003), and aerobic endurance (*p* = 0.005). There were also significant main effects for time on waist circumference (*p* = 0.009), handgrip strength (*p* = 0.004), lower limbs power (*p* < 0.001), reaction time (*p* = 0.034) and left-handed coordination (*p* = 0.040). There were also significant improvements in IG on waist circumference (*p* = 0.008), handgrip strength (*p* = 0.025), flexibility (*p* = 0.014), lower limbs power (*p* < 0.001), aerobic endurance (*p* = 0.003) and reaction time (*p* = 0.028). Participants from CG also improved handgrip strength (*p* = 0.039). Significant differences were also found between IG and CG (IG improved, CG did not) on coordination (*p* = 0.045), core resistance (*p* = 0.040), functional mobility (*p* = 0.049), visceral fat (*p* = 0.036) and running speed and agility (*p* = 0.014) Positive (not statistically significant) results: There was a trend towards significant differences between IG and CG on body weight (*p* = 0.059) Negative results/no changes: There were no significant interactions or main effects for body fat percentage, muscle mass and right-handed coordination
Ayán Pérez et al. [40]	Sample size [n pre/post; sex; age (years)] 14/14; 7 female; 37.07 ± 7.34	IG: 12 weeks of water-based physical exercise training sessions Two 45 min sessions/week in a pool Warm up (15 min): breathing exercises, crawl kicks holding the edge of the pool Main part (30 min): crawl stroke, backstroke Cool down (5 min): ludic activities in a higher temperature pool or in the whirlpool bath	Height (NR) Weight (NR) BMI (NR) Waist circumference (NR) Body composition (triceps and subscapular skinfolds) Cardiorespiratory fitness (20 m shuttle run test) Agility (4 × 10 m shuttle run test) Strength (handgrip test) Power (standing broad jump test) Swimming ability (specific observation scale) Quality of life (specific questionnaire)	IG adherence (%); mean attendance (%): 100%; >85% Positive (not statistically significant) results: IG increased 6 s in the cardiorespiratory fitness test (151.57 ± 66.70 vs. 157.50 ± 64.89) and reduced approximately 3 s in the agility test (30.16 ± 7.20 vs. 27.75 ± 5.20) Negative results/no changes: No significant changes were observed for any outcome
Mohamed et al. [41]	Sample size (n pre/post; sex) 45/45; 24 female Distribution: group; sex; age (years) IG1: n = 15; 8 girls; 10.86 ± 0.89 IG2: n = 15; 10 girls; 11.06 ± 0.84 CG: n = 15; 6 girls; 11.3 ± 0.92	IG1: 12 weeks of aerobic exercise in cycle ergometer + PNF of the respiratory muscles Aerobic exercise in cycle ergometer Five 20 min sessions/week Gradual increase in resistance Warm up: 5 min, low speed Main part: 10 min, increasing resistance Cool down: 5 min, unloaded cycling PNF training Five 20 min sessions/week Three stages: diaphragmatic stimulation, stimulation of upper lateral costal regions, stimulation of lower lateral costal regions 3 min break between stages Stabilization reversal technique IG2: 12 weeks of aerobic exercise in cycle ergometer + IMT Aerobic exercise in cycle ergometer: described in IG1 IMT training Five 20 min sessions/week Threshold-loading IMT device provided constant resistance in each inspiration First session 20% of MIP, rest of the intervention 40% CG: 12 weeks of aerobic exercise in cycle ergometer (described in IG1)	Respiratory muscle strength—MIP and MEP (respiratory pressure meter) Ratio of the upper to lower chest wall (anteroposterior chest radiograph) Pulmonary function—vital capacity, forced expiratory volume in 1 s, peak expiratory flow rate, maximum voluntary ventilation (spirometry) Aerobic endurance (6-minute walk test)	IG adherence (%); mean attendance (%): 100%; NR Significant results: There was a significant interaction of treatment and time (Wilks’ Lambda = 0.06; F (16, 70) = 13.27, *p* = 0.001, h2 = 0.75). There also was a significant main effect of time (Wilks’ Lambda = 0.02; F (8, 35) = 157.28, *p* = 0.001, h2 = 0.97) and treatment (Wilks’ Lambda = 0.17; F (16, 70) = 6, *p* = 0.001, h2 = 0.57). IG1 and IG2 showed an increase in all outcomes post-test (*p* < 0.001), while CG only increased MIP, MEP, ratio of upper to lower chest wall, vital capacity, peak expiratory flow rate, and aerobic endurance (*p* < 0.05). Mean differences between pre and post treatment were greater than MCID in IG1 and IG2, and smaller in CG. There was a significant increase in all outcomes in IG1 compared with IG2 (*p* < 0.05) and CG (*p* < 0.001), and in IG2 compared with CG (*p* < 0.05)
Ringenbach et al. [42]	Sample size [n pre/post; sex; age (years)] 14/14; 6 female; 26.25 ± 5.17	Block randomization into three different training sessions 4 total sessions (first session consisted on pre-test assessment) IG1: Resistance training session Warm up: 5 min, dynamic joint mobility Main part: 30 min, 6 exercises in weight-stack machines, 2 sets of 8–12 repetitions at 75% of 1RM IG2: Assisted cycling therapy Warm up: 5 min, cycling at one’s own pace Main part: 20 min, 135% of baseline voluntary speed CG: No training 35 min of simple board game play	Heart rate (heart rate device) Inhibition control (Eriksen Flanker Task test) Cognitive planning (Tower of London test)	IG adherence (%); mean attendance (%): 100%; 100% Significant results: For cognitive planning, there was a significant interaction between intervention and time [F(2,20) = 3.08, *p* = 0.034]. There also was a main effect of time [t(10) = −1.99, *p* = 0.038] in CG Positive (not statistically significant) results: For inhibition control, there was a trend towards significance for the main effect of time [F(1,12) = 1.062, *p* = 0.16] on the percent correct responses in all interventions, and for the main effect of time [t(13) = −1.1, *p* = 0.15] on inhibition time in IG2. There was also a non-significant reduction in inhibition time on IG1 Negative results/no changes: Inhibition time increased in CG. No differences found on cognitive planning in IG1 and IG2.
Camacho et al. [43]	Sample size [n pre/post; sex; age (years)] 39/39 adults with DS; 15 female; 29 ± 3 39 informants (parents/teachers) Distribution (adults with DS): IG: n = 9 CG: n = 30	IG: Two physical education sessions/week + two 90 min soccer training sessions/week Physical education sessions Circuits involving basic movement patterns, basic physical qualities, and adapted sports practice Soccer training sessions Warm up: group games for general activation and specific motor activities Main part: specific technical and tactical soccer drills, real play situations Cool down: stretching CG: Two physical education sessions/week (described in IG)	Quality of life—self-determination, rights, emotional well-being, material well-being, physical well-being, social inclusion, interpersonal relationships, personal development, and Quality of Life Index (specific questionnaire)	IG adherence (%); mean attendance (%): 100%; NR Significant results: A statistically significant correlation (r = −0.353; *p* = 0.027) with moderate magnitude and negative meaning between age and IG results on physical well-being. IG perceptions showed significant sex differences (men > women) on emotional well-being (Z = −2.29; *p* = 0.022), material well-being (Z = −2.29; *p* = 0.022), and personal development (Z = −2.20; *p* = 0.028), and group differences (IG > CG) in all outcomes (*p* < 0.001). Informants perceptions also indicated significant differences between genders (men > women) on social inclusion (Z = −2.49; *p* = 0.013), emotional well-being (Z = −2.29; *p* = 0.022), physical well-being (Z = −2.45; *p* = 0.014), material well-being (Z = −3.88; *p* < 0.001), and Quality of Life Index (Z = −2.84; *p* = 0.004). Results of IG were significantly higher that informants results on social inclusion (Z =−2.89; *p* = 0.004), self-determination (Z = −4.25; *p* = 0.001), material well-being (Z = −2.88; *p* = 0.004), personal development (Z = −2.39; *p* = 0.017), and Quality of Life Index (Z = −3.27; *p* = 0.001) Negative results/no changes: No correlations between age and IG or informants results on any outcome (except for physical well-being). No sex differences were detected in IG perceptions on self-determination, rights, physical well-being, social inclusion, interpersonal relationships, and Quality of Life Index, and in informants perceptions on self-determination, rights, interpersonal relationships, and personal development. No group differences were found for any outcome in informants perceptions. There was no difference between IG and informants perceptions on rights, emotional well-being, physical well-being, and interpersonal relationships
Di Fabrizio et al. [44]	Sample size [n pre/post; sex; age (years)] 10/10; 3 female; 26.5 ± 7.59	IG: 10 months of adapted physical activity protocols Three sessions/week Different types of exercise: proprioception, coordination, stretching, balance, and muscle strengthening	Posture (computerized videography) Baropodometry—plantar surface, pressure, force, and load distribution (baropodometric platform)	IG adherence (%); mean attendance (%): 100%; NR Positive (not statistically significant) results: Results showed a reduction in podalic overload points and a better distribution of the podalic load on both feet, both as regards the ratio of both feet and as regards the distribution between forefoot and rearfoot. Seven participants also improved in terms of joint degrees, position of the hip, knee, ankle, and foot joints, as well as a reduced pelvic anteversion and knee and rearfoot valgism, and an elevation of the plantar vault
Hojlo et al. [45]	Sample size (n pre/post; sex) 13/12; 8 female Distribution: group; sex; age (years) IG1: n = 7; 4 girls; 14–16 IG2: n = 7; 4 girls; 11–17 2 participants repeated from IG1 to IG2	Two iterations of DSFit, a structured group exercise program for adolescents with DS with weekly meetings and independent home exercise sessions Exercises addressed core/trunk strength and stability, lower- and upper-body strength, balance, flexibility, and walking After sharing the planning of the session, previous exercises were reviewed and practiced; after that, new exercises were introduced. Then, a 10 min walk/dance break was included. Finally, all learned exercises were practiced in one sequence Participants were given a visual guide of the new exercises learned, and a paper log to record exercises at home IG1: 10 weeks and 8 total sessions IG2: 8 weeks and 7 total sessions	Height (stadiometer) Weight (scale) BMI (NR) Muscular strength and endurance (overhead/wall squat test, Sit to Stand test, modified Push-Up test, bird dog time, Trunk Lift test, plank time, Curl Up test, dynamometry) Flexibility (shoulder stretch, Sit and Reach test) Balance and gait (Timed Up and Go test) Aerobic endurance (6-min Walk Test) Goal setting (specific final survey) Feedback (specific final survey) Anxiety and depression (specific questionnaire) Hyperactivity and impulsivity (specific questionnaire)	IG adherence (%); mean attendance (%): 92%; NR Positive (not statistically significant) results: Repeating participants data reported better healthy habits related to exercising, lower BMI and, in one of them, higher motivation and self-confidence. The majority of participants in IG1 and IG2 improved in at least one physical outcome. Anxiety and depression decreased or remained the same for the majority of participants, and hyperactivity and impulsivity decreased for almost all of them. Participants and parents started to set physical activity and health goals in their daily lives, acknowledging the importance of a healthy lifestyle; they also identified fun, a positive and social climate, meeting other families, and practicing walking, ball or dance/musical exercises as key aspects for participant satisfaction. Parents also perceived an improvement in independence of participants; the latter also expressed enjoyment in learning new exercises and being relational with other children Negative results/no changes: Height, weight, and resting heart rate did not significantly change for any cohort. Waiting and standing exercises were perceived by participants as dissatisfying
Perić et al. [46]	Sample size (n pre/post; sex) 25/25; sex NR Distribution: group; age (years) IG: n = 12; 15.68 ± 0.49 CG: n = 13; 15.72 ± 0.46	IG: Usual daily regime (no physical activity) + 16 weeks of adapted soccer program Two 60 min sessions/week Warm up (10 min): running and shuttle run with ball Main part (45 min): Exercises in pairs (first three weeks) and in threes (13 remaining weeks) working on basic elements of football (dribbling, passing, receiving, goal kicking, double pass, cooperation with other players); competitive tasks (after eight week; 3 vs. 3, 4 vs. 4) in the last 15 min of the main part Cool down (5 min): stretching CG: Usual daily regime (no physical activity)	Aggression (specific observation scale) Attention disorders (specific observation scale) Anxiety and depression (specific observation scale) Social problems (specific observation scale) Soccer skills (Special Olympics soccer skills test)	IG adherence (%); mean attendance (%): 100%; 100% Significant results: Results showed a significant interaction effect of factors (time and group) for all psychosocial outcomes (aggression, Wilks’ Lambda = 0.501, F = 17.930, *p* < 0.001, ηp2 = 0.499; attention disorders, Wilks’ Lambda = 0.507, F = 17.519, *p* = 0.001, ηp2 = 0.493; anxiety and depression, Wilks’ Lambda = 0.518, F = 16.761, *p* = 0.001, ηp2 = 0.482; social problems, Wilks’ Lambda = 0.584, F = 12.800, *p* = 0.002, ηp2 = 0.416) and straight dribbling (Wilks’ Lambda = 0.278, F = 59.800, *p* < 0.001, ηp2 = 0.722). Significant improvements (*p* < 0.05) were also found in IG for all psychosocial outcomes and only for one soccer skill (straight dribbling). No changes were detected in CG for any outcome
Benavides Pando et al. [47]	Sample size [n pre/post; sex; age (years)] 26/18; 10 female; 17.44 ± 4.26	IG: 10 weeks of adapted physical activity program based on cheerleading Two 30/45 min sessions/week Warm up: NR Main part: Exercises focused on muscular endurance, aerobic endurance, coordination, balance, and flexibility	Strength (Handgrip test) Power (Standing Broad Jump test) Coordination (Plate Tapping test) Flexibility (Sit and Reach test) Running speed and agility (10 × 5 m Shuttle Run test) Muscular endurance (30-s Sit Up test) Balance (Flamingo test)	IG adherence (%); mean attendance (%): 69%; NR Significant results: Improvements in core resistance (*p* = 0.002), power in lower limbs (*p* = 0.021), coordination in right upper limb (*p* = 0.01), flexibility (*p* < 0.001), and running speed and agility (*p* < 0.001) Negative results/no changes: No changes in coordination in left upper limb, balance, and handgrip strength (all *p* > 0.05)

Note. 1RM, one repetition maximum; BMI, body mass index; CG, control group; h, hour; IG, intervention group; IMT, inspiratory muscle training; MCID, minimal clinically important difference; MEP, maximal expiratory pressure; min, minutes; MIP, maximal inspiratory pressure; MVPA, moderate-to-vigorous physical activity; NR, not reported; PNF, proprioceptive neuromuscular facilitation; s, seconds; WBV, whole body vibration.

## Data Availability

The data presented in this study are available on request from the corresponding author.

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
