# Peer review of "Impact of Physical Activity on Autonomy and Quality of Life in Individuals with Down Syndrome: A Systematic Review"

_healthcare, 2024, doi:10.3390/healthcare12020181_

Round 1
Reviewer 1 Report
Comments and Suggestions for Authors
The authors must be commended for carrying out a study regarding the impact of physical activity on individuals with Down Syndrome. This topic is very important, the research methodology used in the study is appropriate, and the manuscript is written with great clarity. However, some issues need to be taken into consideration. My biggest remark is related to the absence of meta-analysis. Since there are not many studies on this topic, I think that meta-analysis is applicable, and it would give an outstanding contribution to this great study. My specific comments are presented in the following text.
Abstract
Line 23: Please avoid starting a sentence with numerals.
Introduction
Section 3.1, the first and the second sentence: Please add references after this kind of statement.
I generally think that you should consider expanding the term ‘’physical activity’’. What physical activity? I suppose you are referring to planned, systematic and regular physical activity…
Results and discussion
From my point of view, the result section is too long and the discussion section is too short. Namely, I think that most of the results part belong to the discussion section; the discussion about the conducted studies on this topic (which is the essence of the review study) logically belongs to the discussion section.
Author Response
Dear reviewer,
Thank you for your kind words and your suggestion about this research, we are glad to hear that other experienced researchers find this study useful. We will try to answer all your comments and concerns about our manuscript, quoting your words and indicating the changes made and the location in the revised manuscript (or the reason why we did not make any modification). The specific lines will be included regarding the pdf document. You can see all the changes made in the manuscript in the Microsoft Word document, they have been made with track changes option.
- My biggest remark is related to the absence of meta-analysis. Since there are not many studies on this topic, I think that meta-analysis is applicable, and it would give an outstanding contribution to this great study. My specific comments are presented in the following text.
Thank you for your consideration about the importance of meta-analysis in this field of study. We did not carry out a meta-analysis in this review because of the clinical heterogeneity of study designs (i.e., different types of physical activity interventions, inclusion of non-randomized studies), characteristics of the samples (i.e., different ages and experiences with physical activity), variables analyzed (i.e., different variables and different measurement instruments in the same variables among studies), and research questions (i.e., different research questions and goals among studies). Methodological characteristics of the selected studies can be found in Tables 2 and 3.
Following the recommendations of Lensen (Lensen, S. (2023). When to pool data in a meta-analysis (and when not to)?. Fertility and Sterility, 119(6), 902-903., https://doi.org/10.1016/j.fertnstert.2023.03.015) and Escrig et al. (Escrig, V., Llueca, J., Granel, L., & Bellver, M. (2021). Meta-Analysis: A Basic Way To Understand And Interpret Your Evidence. Revista de Senologia y Patologia Mamaria, 34(1), 44-51, https://doi.org/10.1016/j.senol.2020.05.007), we believe a meta-analysis is not appropriate for this review.
- Line 23: Please avoid starting a sentence with numerals.
Sentence has been changed to avoid this issue (L23).
- Section 3.1, the first and the second sentence: Please add references after this kind of statement.
The whole Introduction section has been modified to answer the comments of the other reviewer, and this issue has been solved (L44-50).
- I generally think that you should consider expanding the term ‘’physical activity’’. What physical activity? I suppose you are referring to planned, systematic and regular physical activity…
A definition of physical activity has been included (L45-47).
- Results and discussion. From my point of view, the result section is too long and the discussion section is too short. Namely, I think that most of the results part belong to the discussion section; the discussion about the conducted studies on this topic (which is the essence of the review study) logically belongs to the discussion section.
Results and discussion have been changed to expand the latter and to interpret the data more thoroughly.
We hope we have answered all your considerations. Again, thank you very much for your time and effort reviewing our manuscript. Best regards.
Reviewer 2 Report
Comments and Suggestions for Authors
I would like to begin by commending you for your significant contribution to the field of Quality of Life research through your manuscript titled "Impact of Physical Activity on Autonomy and Quality of Life in Individuals with Down Syndrome: a Systematic Review," submitted for publication in Healthcare. The topic of your research is indeed interesting and relevant, and I appreciate the dedication and effort you have invested in this study.
However, after a thorough review, I believe that there are certain areas of your manuscript that require substantial revisions before it can be considered for publication. These revisions, in my view, constitute major changes, and I have outlined my suggestions below:
Introduction:
The Introduction provides a comprehensive description of Down syndrome, but it lacks essential information about the conceptions of quality of life and autonomy. I strongly recommend that you incorporate a more concise and focused discussion of Down syndrome, while dedicating more space to defining and explaining autonomy and quality of life, including their relevant aspects.
Results Section:
In the Results section, lines 200-203, the inclusion of "moderate-to-vigorous physical activity (MVPA)" and "adherence" as aspects related to autonomy raises doubts. If these aspects are indeed relevant to autonomy, it is essential to provide a more detailed explanation or reference in the introduction to clarify their significance.
Lines 341-371 include various physical well-being aspects in the analysis. It remains unclear whether all of these aspects are directly related to quality of life, particularly "Anthropometric Measures" and "Pubertal development." I recommend that you either justify their inclusion in the introduction or reconsider their incorporation into the analysis.
Discussion:
A major revision of the Discussion section is necessary. Currently, it contains extensive repetition of the results section with limited interpretation of the findings. I urge you to focus more on interpreting the results in the context of your research question and provide insights that go beyond summarizing individual studies.
The section where you compare review data with the results of individual studies and other reviews raises a concern. Comparing review data with individual studies may not be meaningful. In the Introduction, you stated that there were no other reviews conducted on this specific topic. However, if similar reviews do exist, it is essential to mention them in the introduction and provide a clear rationale for why your review was considered necessary.
Lastly, your manuscript lacks a discussion of limitations, both regarding the evidence included in the review and the review process itself. I recommend that you explicitly address these limitations in the Discussion section.
Conclusions:
The Conclusions section should be rewritten to avoid duplicating information from the Results and Discussion sections. It should primarily focus on the implications of your findings for practice, policy, and future research.
I believe that addressing these major revisions will significantly enhance the quality and clarity of your manuscript. I look forward to reviewing the revised version of your manuscript once you have had the opportunity to implement these suggestions.
Author Response
Dear reviewer,
Thank you for your kind words and your suggestion about this research, we are glad to hear that other experienced researchers find this study useful. We will try to answer all your comments and concerns about our manuscript, quoting your words and indicating the changes made and the location in the revised manuscript (or the reason why we did not make any modification). The specific lines will be included regarding the pdf document. You can see all the changes made in the manuscript in the Microsoft Word document, they have been made with track changes option.
- The Introduction provides a comprehensive description of Down syndrome, but it lacks essential information about the conceptions of quality of life and autonomy. I strongly recommend that you incorporate a more concise and focused discussion of Down syndrome, while dedicating more space to defining and explaining autonomy and quality of life, including their relevant aspects.
The whole introduction has been restructured. Down Syndrome information has been shortened and a specific section about autonomy and quality of life (section 1.3, L56-77) has been included.
- In the Results section, lines 200-203, the inclusion of "moderate-to-vigorous physical activity (MVPA)" and "adherence" as aspects related to autonomy raises doubts. If these aspects are indeed relevant to autonomy, it is essential to provide a more detailed explanation or reference in the introduction to clarify their significance.
After reflecting on your comment, we have decided to include MVPA-related results within the physical well-being section. In relation to adherence, we have decided to maintain it in autonomy because our manuscript refers to adherence as the ability to keep practicing physical activity autonomously.
- Lines 341-371 include various physical well-being aspects in the analysis. It remains unclear whether all of these aspects are directly related to quality of life, particularly "Anthropometric Measures" and "Pubertal development." I recommend that you either justify their inclusion in the introduction or reconsider their incorporation into the analysis.
Although some of the research analyzed in the review addressed these two variables, we have opted to delete pubertal development results, since they are not really related to the aim of this work. However, within anthropometric measures you can find aspects directly related to obesity and physical health, such as waist circumference or body composition. Therefore, we believe that anthropometric measures should be included in the review.
- A major revision of the Discussion section is necessary. Currently, it contains extensive repetition of the results section with limited interpretation of the findings. I urge you to focus more on interpreting the results in the context of your research question and provide insights that go beyond summarizing individual studies.
A complete restructuring has been made in the discussion. It has been expanded, trying to interpret the results and to connect them with scientific literature to gain insight into the reason why the reviewed investigations have achieved those results.
- The section where you compare review data with the results of individual studies and other reviews raises a concern. Comparing review data with individual studies may not be meaningful. In the Introduction, you stated that there were no other reviews conducted on this specific topic. However, if similar reviews do exist, it is essential to mention them in the introduction and provide a clear rationale for why your review was considered necessary.
To our knowledge, there was no previous systematic review focused on the effects of physical activity on all dimensions of quality of life in Down syndrome by the time we started the formal search of the review, neither in databases (e.g., WoS, Scopus) or in PROSPERO. There were some obesity- and physical well-being-related reviews carried out in Down Syndrome and in intellectual disabilities in general, but not a single one addressing the whole concept of quality of life.
We made comparisons of the results both with older research that would have been excluded from our review or with research that included different intellectual disabilities in the participants (including Down Syndrome, of course). We tried to address every possibly interesting research that could help to gain insight into the topic.
- Lastly, your manuscript lacks a discussion of limitations, both regarding the evidence included in the review and the review process itself. I recommend that you explicitly address these limitations in the Discussion section.
A specific Limitations section has been included in the discussion (L640-660).
- The Conclusions section should be rewritten to avoid duplicating information from the Results and Discussion sections. It should primarily focus on the implications of your findings for practice, policy, and future research.
Conclusions have been rewritten (L661-695).
We hope we have answered all your considerations. Again, thank you very much for your time and effort reviewing our manuscript. Best regards.
Round 2
Reviewer 1 Report
Comments and Suggestions for Authors
Thank you for taking into consideration all my suggestions. The paper is now dramatically improved and it is suitable for publication, from my point of view. It will be a great contribution to scientific knowledge.
Best wishes
Reviewer 2 Report
Comments and Suggestions for Authors
I am pleased to inform you that after a thorough examination of your revised manuscript, I have found that the revisions have significantly enhanced the overall quality of your manuscript, addressing the concerns raised during the initial review process effectively. As a result of these comprehensive revisions, I have recommended to the editor that your manuscript be accepted for publication in its current form.